# Universal Self-Consistency for Large Language Models

**Xinyun Chen** [* 1]   **Renat Aksitov** [* 1]   **Uri Alon** [1]   **Jie Ren** [1]   **Kefan Xiao** [1]   **Pengcheng Yin** [1]   **Sushant Prakash** [1]   **Charles Sutton** [1]   **Xuezhi Wang** [1]   **Denny Zhou** [1]

## Abstract

Self-consistency with chain-of-thought (CoT) prompting has demonstrated remarkable performance gain by utilizing multiple reasoning paths sampled from large language models (LLMs). However, self-consistency relies on heuristics to extract answers and aggregate multiple solutions, which is not applicable to solving tasks with free-form answers. In this work, we propose Universal Self-Consistency (USC), which leverages LLMs themselves to select the most consistent answer. We evaluate USC on a variety of benchmarks, including mathematical reasoning, code generation, long-context summarization, and open-ended question answering. On open-ended generation tasks where the original self-consistency is not applicable, USC effectively leverages multiple samples and improves the performance. For math reasoning, USC matches the standard self-consistency performance without requiring the answer formats to be similar. Finally, without access to execution results, USC also performs on par with execution-based voting methods on code generation.

## 1. Introduction

Large language models (LLMs) have accomplished significant breakthroughs in a wide variety of domains, including mathematical reasoning (Cobbe et al., 2021), code generation (Austin et al., 2021), and other text generation tasks (Bubeck et al., 2023). Despite the rapid progress, LLM-generated responses are still prone to errors when they get long. A long line of efforts have been devoted to improving the output quality by sampling multiple model responses and then selecting the final output based on certain criteria. For example, prior works have trained neural networks to

rerank model outputs (Cobbe et al., 2021; Li et al., 2023b; Ni et al., 2023; Yin & Neubig, 2019), and more recent works investigate using LLMs to score model responses (Fu et al., 2023; Liu et al., 2023; Wang et al., 2023a).

In this work, we consider the *consistency* as the criterion to select model output, a generic metric that has enabled huge performance leaps in reasoning (Wang et al., 2022) and code generation (Li et al., 2022; Shi et al., 2022). The intuition is that if a proposed output appears consistently in many different samples, then the model has high confidence in that output, and so it is more likely to be correct. In particular, *self-consistency* (Wang et al., 2022) selects the final answer as the most common one across samples, essentially marginalizing over the latent reasoning paths produced by chain-of-thought prompting (Wei et al., 2022). This simple method leads to a major boost across a wide variety of benchmarks. However, self-consistency can only be applied to tasks where the final answer can be aggregated via exact match, e.g., a multiple-choice answer, or a single number for math problems. Despite the power of self-consistency, this limitation prevents us from applying it to tasks that require free-form generation, such as code generation, abstractive summarization, and open-end question answering.

To address this major limitation of self-consistency, we propose Universal Self-Consistency (USC) to support various applications, especially free-form generation tasks. Given multiple candidate responses, USC calls the LLM to select the most consistent response as the final output. Thus, USC eliminates the need of designing an answer extraction process, and is applicable to tasks with free-form answers. Although prior works have revealed weaknesses of LLMs for response selection, such as position bias (Wang et al., 2023b; Zheng et al., 2023b) and incorrectly judging the answer correctness (Huang et al., 2023b; Gou et al., 2023), we find that USC does not suffer from them. Intuitively, assessing the consistency among candidate answers is easier for an LLM than judging the answer quality directly.

We evaluate universal self-consistency on a wide range of tasks, including open-ended question answering, long-context summarization, code generation and mathematical reasoning. We first show that USC improves the performance for open-ended question answering (Lin et al.,

---
*Equal contribution   [1]Google LLC. Correspondence to: Xinyun Chen <xinyunchen@google.com>, Renat Aksitov <raksitov@google.com>, Denny Zhou <dennyzhou@google.com>.

*Proceedings of the 1st Workshop on In-Context Learning at the 41st International Conference on Machine Learning*, Vienna, Austria. 2024. Copyright 2024 by the author(s).

2021) and summarization (Huang et al., 2021; Chen et al., 2022b), where the standard self-consistency is not applicable. On programming tasks including text-to-SQL generation (Li et al., 2023a) and Python data science code generation (Yin et al., 2023), USC matches the performance of execution-based consistency (Li et al., 2022; Shi et al., 2022), while USC does not require execution results to perform selection. Finally, on tasks for which standard self-consistency applies, namely GSM8K (Cobbe et al., 2021) and MATH (Hendrycks et al., 2021) benchmarks for math problem solving, USC generally matches the performance of self-consistency. In addition to the performance gain, we also demonstrate that USC outputs highly match those of the standard self-consistency when the comparison is applicable, while it is robust to different response orderings.

## 2. Background: Self-Consistency

Self-consistency (Wang et al., 2022) augments chain-of-thought prompting (Wei et al., 2022) by sampling multiple reasoning chains and then taking a majority vote on the final answer set. The intuition is that sometimes the greedily decoded reasoning process might not be the optimal one, hence it makes more sense to sample a diverse set of reasoning chains, and if some of them lead to the same answer, then we have a higher confidence that this consistent answer is the correct one. It has been shown that self-consistency improves the greedy chain-of-thought prompting by a large margin on a wide set of reasoning tasks.

Besides question answering tasks, consistency-based answer selection has also been applied to code generation (Shi et al., 2022; Li et al., 2022; Chen et al., 2019), which requires code execution. Specifically, we first execute all predicted programs, then programs with the same execution outputs are clustered together, assuming that they are semantically equivalent. Finally, we select the program belonging to the largest cluster as the final prediction. When the program inputs given in the task description are insufficient to distinguish between different predictions, this execution-based code selection is also often accompanied with a test case generation process to better examine the consistency (Li et al., 2022; Chen et al., 2022a; Huang et al., 2023a).

Despite the remarkable improvement, self-consistency is only applicable to problems with a unique and closed-form answer, e.g., when the final answer consists of a single number, because a majority vote needs to be taken over the final answer set. This requirement poses a challenge for tasks that require open-ended generations, such as summarization, creative writing, and open-ended question answering.

## 3. Universal Self-Consistency

We present the overall workflow of universal self-consistency (USC) in Figure 1, which utilizes LLMs to enable self-consistency for a wide variety of tasks, especially free-form text generation. First, we sample multiple responses with the large language model. Afterward, to select one model response as the final answer, we concatenate all responses together, and then construct a prompt with an instruction asking the language model to select the most consistent response. In this way, USC obviates the necessity of counting the exact answer frequency as in the standard self-consistency, and relies on the LLM's own ability to measure the consistency among different responses. Although prior works show that LLMs sometimes have trouble evaluating the prediction correctness (Huang et al., 2023b; Gou et al., 2023), especially for reasoning problems, empirically we observe that LLMs are generally able to examine the response consistency across multiple tasks.

Consistency assessment with LLMs offers more flexibility for free-form generation. Figure 2 demonstrates example tasks where different consistency criteria are beneficial for response selection. Specifically, Figure 2a shows model responses for a math problem with diverse output formats, which are challenging for rule-based methods to extract answers. Nonetheless, assuming that the final answers are correctly extracted, the consistency criterion still follows the standard self-consistency on mathematical reasoning, which is based on the exact match of the final answers represented as single numerical values. On the other hand, Figure 2b shows an example question where the final answer is an entity list. Despite that there is no response that is consistent with others based on the exact match, the LLM selects the response where each of the predicted entities appears most frequently among the candidate outputs. In Section 4, we further show that LLMs can also examine the consistency among responses beyond the question answering tasks, including code generation without access to the execution outputs, and long-context summarization.

## 4. Experiments

### 4.1. Evaluation Setup

**Benchmarks.** We evaluate USC on the following tasks:

- *Mathematical reasoning benchmarks*, including GSM8K (Cobbe et al., 2021) with grade school math word problems, and MATH (Hendrycks et al., 2021) with problems from high school competitions.
- *Code generation benchmarks*, including BIRD-SQL dataset (Li et al., 2023a) for text-to-SQL generation, and ARCADE dataset (Yin et al., 2023) for Python code generation in data science notebooks.
- *Long-context summarization*, including GovReport and SummScreen from ZeroSCROLLS (Shaham et al., 2023). In GovReport (Huang et al., 2021), each input is a document containing ~7,900 words on average,

## Universal Self Consistency prompt

**Responses**

I have generated the following responses to the question {question}
Response 0: {response_0}
Response 1: {response_1}
...
**Select the most consistent response based on majority consensus.**

Question → **LLM** → → **LLM** → **Selected response**

Figure 1: Overview of the Universal Self-Consistency workflow.

and the reference output is an expert-written executive summary with ∼500 words. In SummScreen (Chen et al., 2022b), every input is a transcript of a TV show episode with ∼5,600 words, and each reference output is a ∼100 words human-written recap of the episode. We follow Shaham et al. (2023) and measure ROUGE 1, ROUGE 2, and ROUGE-Lsum which measure n-gram overlap with the reference summary, and we also measure BERTScore F1 (Zhang et al., 2019).

- *TruthfulQA (Lin et al., 2021) benchmark* for open-ended question answering, which contains 817 questions to test model's ability in generating truthful answers. To evaluate the answer's quality, we use the GPT-judge and GPT-info, which are GPT-3 models fine-tuned on human feedback data, provided by (Lin et al., 2021). GPT-judge model outputs a binary rating for truthfulness, and GPT-info model outputs a binary rating for informativeness.

**Decoding schemes.** We compare USC to the following decoding schemes:

- *Greedy decoding* generates a single answer with the temperature 0.
- *Random* selects one answer randomly from multiple samples with temperature > 0.
- *SC* (Wang et al., 2022) is the standard self-consistency decoding with answer extraction. We evaluate SC whenever applicable.

To perform a fair comparison, when sampling (i.e., except in greedy decoding), we always select the final answer from the same set of initial model responses. In code generation, we compare our approach to execution-based self-consistency (Shi et al., 2022; Li et al., 2022; Chen et al., 2019), where we select the code with the most common execution output. Both USC and execution-based self-consistency first filter out syntactically invalid candidates, and perform voting over the remaining programs. In the ARCADE benchmark, we also evaluate a variant of execution-based self-consistency with fuzzy matching as described in Yin et al. (2023), which uses heuristics to de-

termine whether the execution outputs of two programs are equivalent when they are not exact match.

**Implementation details.** We conduct experiments using instruction-tuned `PaLM 2-L` (Anil et al., 2023) and `gpt-3.5-turbo` models. For `gpt-3.5-turbo`, we used the version `gpt-3.5-turbo-0613`. Unless otherwise specified, the LLM generates 8 initial samples for both SC and USC. For mathematical reasoning, summarization and the ARCADE benchmark for Python code generation, the initial samples are generated with zero-shot prompting, thus the output formats are diverse. For the standard self-consistency, we employ a regular expression matching to extract the final answer on GSM8K, and re-use the answer parsing code from (Zheng et al., 2023a) for MATH. For BIRD-SQL, we used the 1-shot chain-of-thought prompt in Li et al. (2023a), which improves the performance. We also utilized a one-shot prompt for TruthfulQA to improve the quality of candidate responses. We set the temperature to be 0.6 for `PaLM 2-L`, and 1.0 for `gpt-3.5-turbo`.

### 4.2. Main Results

**TruthfulQA.** Table 1 presents results on TruthfulQA, where SC is also not directly applicable because the generated answers are in free-form. Comparing with greedy decoding and random selection, USC-based answers have the highest truthfulness with both `PaLM 2-L` and `gpt-3.5-turbo`. For informativeness which is considered as a secondary objective, USC-based answers have the highest score on `PaLM 2-L` and the second highest score (0.1 lower than the highest) on `gpt-3.5-turbo`. Considering that GPT-judge and GPT-info models have generally 90-95% validation accuracy on rating prediction (Lin et al., 2021), the 0.1 difference is not considered significant.

**Summarization.** Results for the summarization benchmarks are shown in Table 2. Since the generated summaries are in free-form, the standard self-consistency is not applicable. In GovReport, USC consistently improves over the baselines across all metrics. In Section 4.3, we further show that asking the model to choose the *most detailed* summary

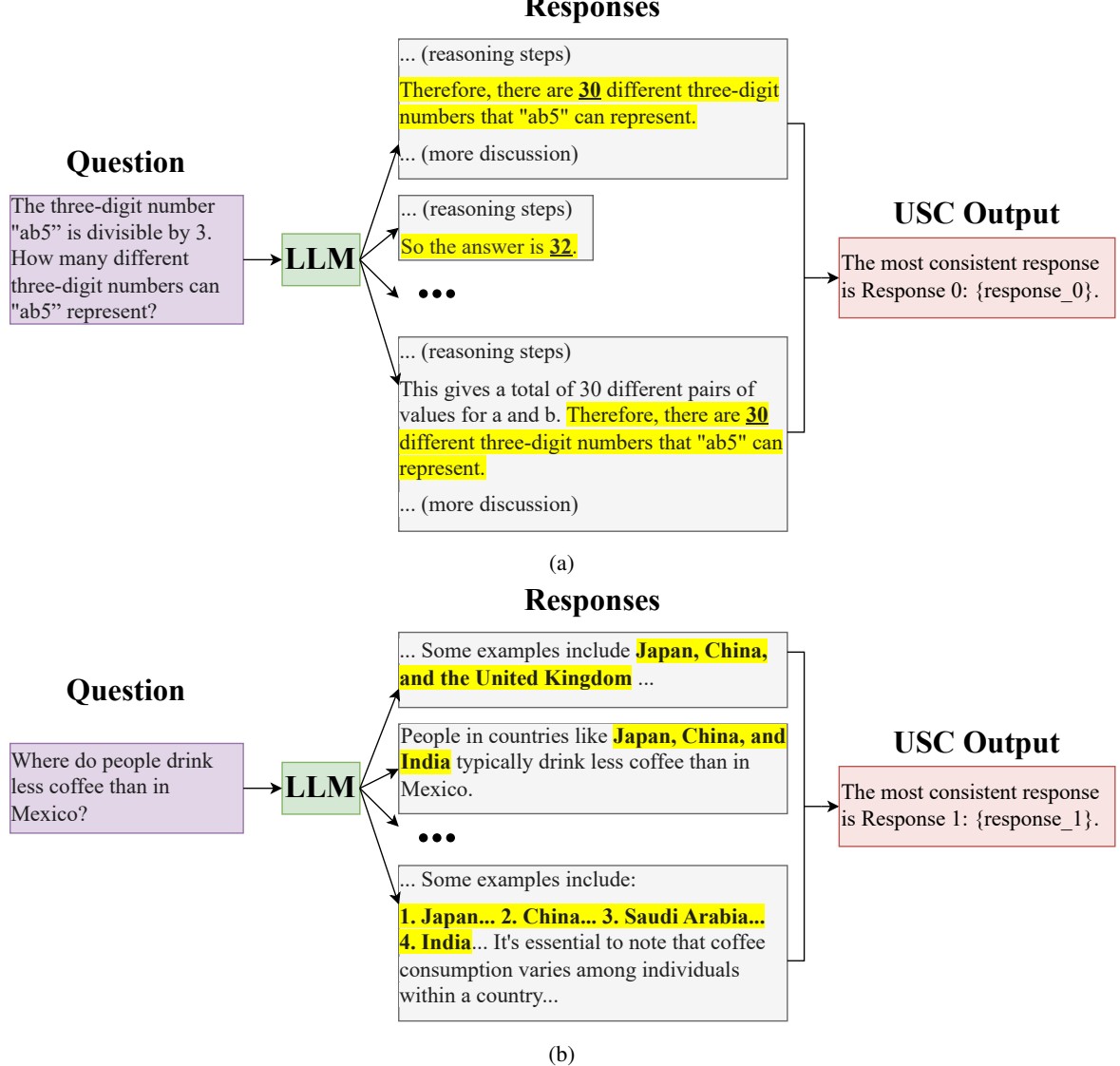

Figure 2: Examples of Universal Self-Consistency for answer selection from responses of diverse formats: (a) mathematical reasoning; and (b) open-ended question answering.

results in an even higher performance gain.

**Code generation.** Table 3 presents the results on BIRD-SQL and ARCADE. On BIRD-SQL, we follow Li et al. (2023a) to also evaluate the valid efficiency score, which measures the efficiency of the generated SQL queries. We show that USC matches the execution-based self-consistency performance on both benchmarks, while USC does not utilize code execution to perform the voting.

**Mathematical reasoning.** We compare USC against the standard self-consistency in Table 4. Overall, USC performance is comparable to the standard self-consistency, which USC does not need answer parsing to perform the voting.

### 4.3. Ablations

**Effect of response ordering.** Prior works have shown that large language models can be affected by the order of candidate responses when used to evaluate their quality (Wang et al., 2023b; Zheng et al., 2023b). From Table 5, we observe that the overall model performance remains similar with different response orders and the variance is low, suggesting the effect of response order is minimal.

**Impact of the number of candidate responses.** Next, we examine the effect of using different numbers of responses in USC. As shown in Figure 3, USC consistently benefits from more samples on TruthFulQA and BIRD-SQL. However, USC does not further improve the performance on

Table 1: Accuracy on TruthfulQA. Since the answer is in free-form, standard self-consistency is not applicable. USC overall has the highest truthfulness and informativeness over the baselines.

| Model | Approach | GPT-judge | GPT-info |
|-------|----------|-----------|----------|
| `PaLM 2-L` | Greedy decoding | 62.1 | 95.1 |
| | Random | 62.9 | 94.6 |
| | USC | **67.7** | **99.0** |
| `gpt-3.5-turbo` | Greedy decoding | 79.8 | **99.7** |
| | Random | 80.6 | 99.3 |
| | USC | **82.5** | 99.6 |

Table 2: Results on long-context summarization benchmarks with `PaLM 2-L`. Since the outputs are in free-form, the standard self-consistency is not applicable. USC consistently improves over the baselines on summary quality.

| Dataset | Approach | ROUGE-1 | ROUGE-2 | ROUGE-Lsum | BERTScore |
|---------|----------|---------|---------|------------|-----------|
| GovReport | Greedy decoding | 38.8 | 16.9 | 33.8 | 62.7 |
| | Random | 38.5 | 16.9 | 33.6 | 62.6 |
| | USC | **40.2** | **17.4** | **35.1** | **62.8** |
| SummScreen | Greedy decoding | 30.6 | 7.5 | 19.1 | **58.7** |
| | Random | 30.2 | 7.3 | 19.0 | 58.6 |
| | USC | **31.7** | **7.8** | **19.8** | 58.3 |

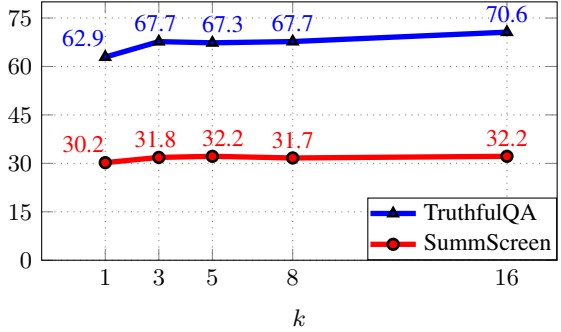

(a) Results in open-ended generation.

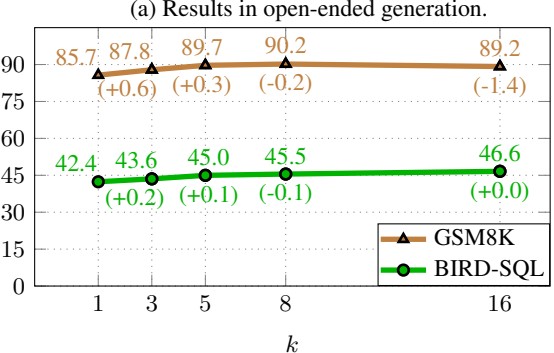

(b) Results in GSM8K and BIRD-SQL. The top numbers are USC accuracies, and the bottom numbers are the differences to SC results.

Figure 3: USC results with different number of samples.

SummScreen after 5 samples, and the accuracy on GSM8K decreases with 16 samples. This can be due to the weakness in long-context understanding when the prompt contains more candidate responses, and the imperfect counting ability of LLMs. Nevertheless, we consider utilizing a few samples (e.g., 8) a sweet spot to balance the task accuracy and compute cost, in which case USC reliably improves the performance across the board. In Section 4.4, we further compare the predictions from USC and SC to understand how using more candidate responses affects the results.

**Criteria for response selection.** One advantage of USC is its generality: the same criteria can be applied to various tasks. Nonetheless, a minor task-specific adaptation of the response selection instruction can further boost the USC performance. For example, Table 6 shows that asking the LLM to choose the most *detailed* response results in gains of about 2 ROUGE-1 and ROUGE-Lsum points.

### 4.4. How well does USC match SC selection?

We have demonstrated that on tasks where the standard self-consistency is applicable, USC and SC achieve comparable overall performance with 8 samples; however, USC fails to further improve the GSM8K performance with 16 samples. In this section, we look closer into the relationship between USC and SC, specifically how well is the alignment between their selected responses.

Figure 4 presents a breakdown analysis of USC predictions

Table 3: Accuracy on code generation benchmarks with `gpt-3.5-turbo`.

| Dataset | Approach | Execution Accuracy | Valid Efficiency Score |
|---------|----------|--------------------|------------------------|
| BIRD-SQL | Greedy decoding | 42.4 | 44.4 |
|  | Random | 41.9 | 44.0 |
|  | SC-Exec | **45.6** | 48.1 |
|  | USC | 45.5 | **48.8** |
| ARCADE | Greedy decoding | 26.0 | |
|  | Random | 26.8 | |
|  | SC-Exec (strict match) | 29.8 | N/A |
|  | SC-Exec (fuzzy match) | **30.3** | |
|  | USC | 30.1 | |

Table 4: Accuracy on mathematical reasoning benchmarks. USC performance is generally comparable to SC.

| Model | Approach | GSM8K | MATH |
|-------|----------|-------|------|
| `PaLM 2-L` | Greedy decoding | 85.7 | 30.8 |
|  | Random | 82.9 | 28.0 |
|  | SC (Wang et al., 2022) | **90.4** | **37.9** |
|  | USC | 90.2 | 37.4 |
| `gpt-3.5-turbo` | Greedy decoding | 73.4 | 33.2 |
|  | Random | 68.5 | 26.3 |
|  | SC | **78.5** | 38.0 |
|  | USC | 77.8 | **38.1** |

Table 5: USC performance with 5 random orders of original responses using `PaLM 2-L`.

(a)

| Dataset | Acc |
|---------|-----|
| GSM8K | $89.7 \pm 0.3$ |
| MATH | $37.3 \pm 0.2$ |

(b)

| Dataset | ROUGE-1 | ROUGE-Lsum |
|---------|---------|------------|
| SummScreen | $31.6 \pm 0.3$ | $19.5 \pm 0.2$ |
| GovReport | $40.0 \pm 0.1$ | $34.9 \pm 0.2$ |

(c)

| Metric | TruthfulQA |
|--------|------------|
| GPT-judge | $68.3 \pm 0.6$ |
| GPT-info | $99.0 \pm 0.1$ |

on mathematical reasoning benchmarks, and Figure 5 further compares the performance of USC and SC when they select different responses. We observe that:

- The voting ties constitute a notable portion to the selection differences between USC and SC, especially with 8 candidate responses. Specifically, among all responses with the maximum votes, SC always selects the one with the smallest index, while USC can pick up alternative ones based on the response format.
- The match ratio between USC and SC consistently surpasses their own accuracies, which shows that consistency is easier to measure than the answer correctness.
- Shifting from 8 to 16 samples, the USC-SC match ratio reduces, suggesting that USC behaves as an imperfect approximation of SC. However, the difference in response selection does not always lead to the performance decrease, as USC sometimes selects the correct

response when SC fails.

## 5. Related Work

**Response reranking and selection for language models.** Reranking is a common method to improve the generation quality in language models by sampling multiple outputs and applying a post-hoc criterion to rank them, which often requires an additional trained ranker and sometimes additional human labeled data. For example, Cobbe et al. (2021) use human labels to train a ranking model to verify whether each generated response is correct or not, and Shen et al. (2021) jointly train a generator and a ranker to improve performance for math tasks. Instead of training response generators and rankers as separate models, Thoppilan et al. (2022) finetune the dialog model to also predict the ratings of candidate responses with human-annotated judgements. For code generation, various reranker models

Table 6: Ablation on the response selection criterion in summarization with `PaLM 2-L`.

| Dataset | Approach | ROUGE-1 | ROUGE-2 | ROUGE-Lsum | BERTScore |
|---------|----------|---------|---------|------------|-----------|
| GovReport | USC | 40.2 | 17.4 | 35.1 | 62.8 |
| | USC – most detailed | **42.4** | **18.2** | **36.9** | **63.2** |
| SummScreen | USC | 31.7 | 7.8 | 19.8 | **58.3** |
| | USC – most detailed | **33.0** | **7.9** | **22.0** | **58.3** |

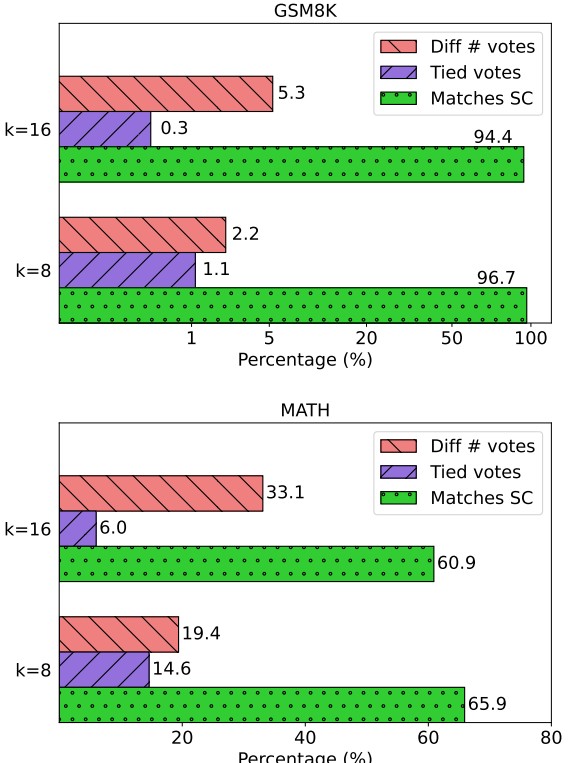

Figure 4: Comparison of selections made by USC versus SC with `PaLM 2-L`. $k$ denotes the number of responses. "Tied votes" represents the case where the USC and SC select different responses, but both have the maximum votes.

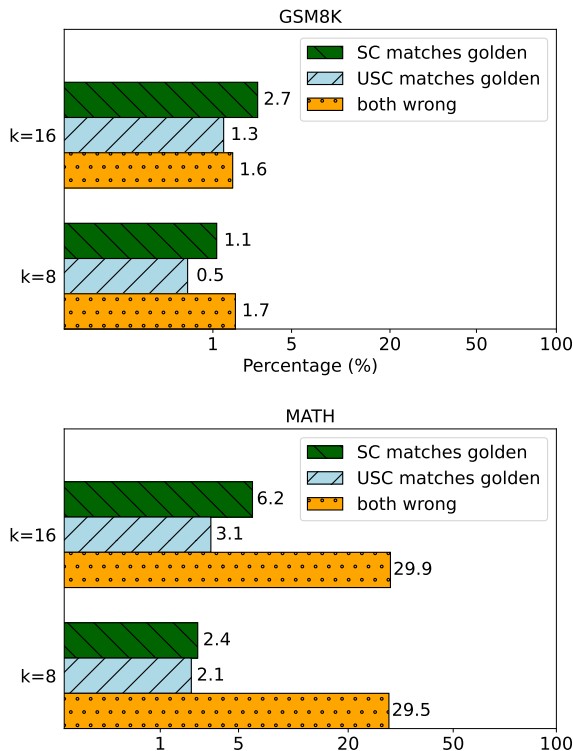

Figure 5: Accuracy distribution when USC selection does not match SC.

have been designed (Ni et al., 2023; Yin & Neubig, 2019; Zeng et al., 2022), which typically utilize execution results and language-specific syntactic features to improve the ranking performance. In contrast with these prior works, USC does not require any additional labeled training data nor an external reranking model: the LLM that generated the initial outputs is the same one that selects the final answer.

Several consistency-based response selection approaches have been proposed in the literature, which typically include a voting procedure to select the most common response (Wang et al., 2022; Zhou et al., 2022; Wightman et al., 2023; Yue et al., 2023; Bertsch et al., 2023). Self-

consistency (Wang et al., 2022) shows that selecting the reasoning path leading to the most common final answer improves the CoT performance. The candidate responses can also come from different prompt variants corresponding to the same problem (Zhou et al., 2022; Wightman et al., 2023; Yue et al., 2023). To measure the pairwise similarity between candidate responses for open-ended generation tasks, Jain et al. (2023) propose the n-gram consistency score, and the consistency score for each response is computed as the sum of the pairwise similarity scores. For code generation, the consistency measurement is typically based on code execution, where the candidate program with the most common execution outputs is selected (Shi et al., 2022; Li et al., 2022; Chen et al., 2019). Besides the consistency of code execution, other works also examine the consistency

between the code and the specification (Min et al., 2023), and utilize it for reranking (Zhang et al., 2023a; Huang et al., 2023a). In this work, we directly instruct the LLM to perform consistency-based selection without an explicit definition of the pairwise similarity, and we demonstrate its applicability to a wide range of tasks.

**Response improvement with multiple candidates.** Some recent works demonstrate that the LLM can improve its prediction output on top of the candidate responses. Yang et al. (2023) show that given a trajectory of previously generated solutions, the LLM can iteratively produce better solutions for an optimization task, and they demonstrate the effectiveness for prompt optimization and several classic mathematical optimization tasks. Other works aggregate multiple reasoning chains and prompts the LLM to generate a better final response, which shows performance improvement on multi-hop question answering (Yoran et al., 2023) and medical question answering (Singhal et al., 2023). Instead of asking the LLM to generate a better response, USC focuses on response selection, as the candidate responses usually already contain high-quality solutions to the underlying tasks. Meanwhile, performing the consistency-based selection is generally an easier task than improving the answer correctness, and we demonstrate that USC properly utilizes multiple responses to improve the performance across different tasks.

**Large language models for response evaluation.** The underlying assumption in our work is that LLMs are reflective enough to evaluate the consistency between multiple responses. This assumption is related to recent findings which had shown that LLMs can be used for evaluating model-generated texts (Bubeck et al., 2023; Fu et al., 2023; Wang et al., 2023a; Zhang et al., 2023b). LLM-based evaluators have demonstrated some promising results, e.g., they can be used to evaluate natural language generations without human references, but some work has also shown that they might not correlate very well with human judgements and sometimes exhibit bias towards model-generated texts (Bubeck et al., 2023; Liu et al., 2023). Another line of work utilizes the prediction probability of the LLM to measure the quality of multiple choices (Ren et al., 2023; Adiwardana et al., 2020), and Lin et al. (2022) show promising results on arithmetic tasks where they prompt the LLM to directly output the level of confidence for its response. In this work, we show that LLMs not only can serve as evaluators, they can also improve their own output by sampling multiple responses and evaluating the consistency between them.

## 6. Limitations and Future Work

Despite that USC supports open-ended generation tasks and generally achieves comparable performance in those domains where the standard self-consistency can be applied, our current USC implementation has its own limitations compared to the extraction-based self-consistency approach.

First, while self-consistency can be applied to an arbitrary number of samples as long as the final answers can be extracted, the number of samples supported by USC is bounded by the context length of the underlying LLM. That said, with the development of more advanced long-context LLMs, the context length is generally sufficient to make best use of the samples.

Second, the voting mechanism in self-consistency inherently offers a measure of confidence or uncertainty for each response (Wang et al., 2022). However, USC has not yet been developed to include the confidence estimation. We consider developing a calibration mechanism for USC as future work, where we can leverage the LLM to perform output clustering and pairwise self-consistency.

Also, USC requires an additional LLM query by design, which incurs additional inference costs. Given that our USC prompt only requires the LLM to generate a response index corresponding to the final answer, the USC output length is much shorter than any individual candidate response to select from. To further reduce the cost, one direction is to use a light-weight language model to conduct USC, and optimizes its efficiency regarding long-context encoding.

Finally, one common limitation of both the standard self-consistency and USC is about the consistency-based selection criterion. Specifically, although consistency is a generic and effective criterion, the most consistent response is not necessarily the best one. In Appendix A, we show that there is still a notable gap to oracle scores where we assume the access to an oracle reranker that always selects the best response. In Section 4.3 we demonstrate that we can design task-specific criteria to further improve the performance, and we consider refining the USC framework to further close the gap to the oracle performance as future work.

## 7. Conclusion

In this work, we present Universal Self-Consistency (USC), which extends the standard self-consistency to support free-form generation tasks. USC notably boosts the performance in diverse applications, and performs on par with the standard self-consistency on those tasks where answer extraction is feasible for voting. Besides addressing the limitations discussed in Section 6, we also consider mitigating the position bias and improving long-context understanding of LLMs as important future work that can further enhance the effectiveness and robustness of the USC scheme.

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

## A. Comparison to Oracle Selection

Tables 7, 8, 9, 10 and 11 compare the results of different approaches to the oracle performance, which selects the best response among candidates for each task. The oracle selection is from the same 8 samples as SC and USC. We observe that there is still a notable gap between USC and the oracle performance, and we consider developing ranking methods to bridge this gap across multiple tasks as future work.

Table 7: Comparison to the oracle selection on mathematical reasoning benchmarks. The results were obtained with `PaLM 2-L`.

| Approach | GSM8K | MATH |
|---|---|---|
| Greedy decoding | 85.7 | 30.8 |
| SC (Wang et al., 2022) | **90.4** | **37.9** |
| USC | 90.2 | 37.4 |
| Oracle | 96.2 | 57.2 |

Table 8: Comparison to the oracle selection on BIRD-SQL benchmark.

| Approach | Execution Accuracy | Valid Efficiency Score |
|---|---|---|
| Greedy decoding | 42.4 | 44.4 |
| SC-Exec | **45.6** | 48.1 |
| USC | 45.5 | **48.8** |
| Oracle | 53.3 | 55.7 |

Table 9: Comparison to the oracle selection on ARCADE benchmark.

| Approach | Execution Accuracy |
|---|---|
| Greedy decoding | 26.0 |
| SC-Exec (strict match) | 29.8 |
| SC-Exec (fuzzy match) | **30.3** |
| USC | 30.1 |
| Oracle | 40.5 |

## B. Examples of USC Prompts

Figures 6 and 7 present examples of full USC prompts with candidate responses for different tasks.

Table 10: Comparison to the oracle selection on long-context summarization benchmarks.

| Dataset | Approach | ROUGE-1 | ROUGE-2 | ROUGE-Lsum | BERTScore |
|---------|----------|---------|---------|------------|-----------|
| GovReport | Greedy decoding | 38.8 | 16.9 | 33.8 | 62.7 |
| | USC | **40.2** | **17.4** | **35.1** | **62.8** |
| | Oracle | 46.1 | 21.3 | 40.3 | 64.7 |
| SummScreen | Greedy decoding | 30.6 | 7.5 | 19.1 | **58.7** |
| | USC | **31.7** | **7.8** | **19.8** | 58.3 |
| | Oracle | 36.9 | 10.8 | 23.6 | 60.6 |

Table 11: Comparison to the oracle selection on TruthfulQA benchmark.

| Model | Approach | GPT-judge | GPT-info |
|-------|----------|-----------|----------|
| PaLM 2-L | Greedy decoding | 62.1 | 95.1 |
| | USC | **67.7** | **99.0** |
| | Oracle | 93.8 | 100.0 |
| gpt-3.5-turbo | Greedy decoding | 79.8 | **99.7** |
| | USC | **82.5** | 99.6 |
| | Oracle | 94.9 | 100.0 |

---

I have generated the following responses to the question: The three-digit number "ab5" is divisible by 3. How many different three-digit numbers can "ab5" represent?

**Response 0:** A number is divisible by 3 if the sum of its digits is divisible by 3. In this case, the sum of the digits of "ab5" must be divisible by 3, so a + b + 5 must be divisible by 3.
The smallest possible sum with a + b + 5 is 1 + 0 + 5 = 6, which is divisible by 3. The largest possible sum is 9 + 7 + 5 = 21, which is also divisible by 3.
Therefore, the possible values of a + b can range from 1 to 16, with each increment of 3 leading to another possible sum. There are 6 such increments (16 - 1) / 3 + 1 = 6, so there are 6 possible values for a + b.
For each possible value of a + b, there are multiple combinations of digits that add up to that value. For example, if a + b = 4, possible combinations are (1, 3), (2, 2), (3, 1) and (4 ,0).
Therefore, there are 30 different three-digit numbers that "ab5" can represent. (1 + 4 + 7 + 9 + 6 + 3 = 30)
In general, any three-digit number of the form "ab5" where a + b + 5 is divisible by 3 will be divisible by 3.
**Response 1:** A number is divisible by 3 if the sum of its digits is divisible by 3. Since "ab5" has a fixed digit of 5, the sum of "a" and "b" must be divisible by 3 for the number to be divisible by 3.
There are 10 possible digits: 0, 1, 2, 3, 4, 5, 6, 7, 8, and 9.
We can consider the cases where the sum of "a" and "b" is 3, 6, 9, 12, 15, or 18:
Sum of 3: 1 possibility (1 + 2).
Sum of 6: 2 possibilities (1 + 5, 2 + 4).
Sum of 9: 3 possibilities (1 + 8, 2 + 7, 3 + 6).
Sum of 12: 4 possibilities (3 + 9, 4 + 8, 5 + 7, 6 + 6).
Sum of 15: 4 possibilities (4 + 11, 5 + 10, 6 + 9, 7 + 8).
Sum of 18: 2 possibilities (9 + 9, 8 + 10).
Therefore, there are a total of 1+2+3+4+4+2 = 16 possibilities for "a" and "b."
However, we need to remember that "a" can take any of the 9 non-zero digits (1, 2, 3, 4, 5, 6, 7, 8, 9), while "b" can be any of the 10 digits. So, we need to double the total number of possibilities for "a" and "b."
Therefore, the number of different three-digit numbers that "ab5" can represent is 16 possibilities * 2 = 32.
So the answer is 32.
**Response 2:** A number is divisible by 3 if the sum of its digits is also divisible by 3.
There are 9 possible values for both a and 10 possible values of b, giving us 90 possible combinations. However, only the combinations where a + b + 5 is a multiple of 3 will be valid.
We can list out the possible combinations:
a + b = 1: (1, 0) a + b = 4: (1, 3), (2, 2), (3, 1), (4, 0) a + b = 7: (1, 6), (2, 5), (3, 4), (4, 3), (5, 2), (6, 1), (7, 0) a + b = 10: (1, 9), (2, 8), (3, 7), (4, 6), (5, 5), (6, 4), (7, 3), (8, 2), (9, 1) a + b = 13: (4, 9), (5, 8), (6, 7), (7, 6), (8, 5), (9, 4) a + b = 16: (7, 9), (8, 8), (9, 7) This gives a total of 30 different pairs of values for a and b. Therefore, there are 30 different three-digit numbers that "ab5" can represent.
1 + 4 + 7 + 9 + 6 + 3 = 30
I hope this helps! Let me know if you have any other questions.
...

Evaluate these responses.
Select the most consistent response based on majority consensus.
Start your answer with "The most consistent response is Response X" (without quotes).

---

Figure 6: A full example USC prompt for mathematical reasoning. This corresponds to the question in Figure 2a.

I have generated the following responses to the question: Where do people drink less coffee than they do in Mexico?

**Response 0:** There are several countries where people drink less coffee than they do in Mexico. Some examples include Japan, China and the United Kingdom.
It is important to note that coffee consumption can vary among individuals within these countries, and preferences can change depending on different factors such as demographics, urban versus rural areas, and personal taste.
**Response 1:** People in countries like Japan, China, and India typically drink less coffee than they do in Mexico.
1. Japan: Tea, particularly green tea, is more popular than coffee in Japan. Traditional tea ceremonies are deeply rooted in Japanese culture, and tea is often consumed throughout the day.
2. China: Tea is also the preferred hot beverage in China. Chinese tea culture dates back thousands of years, and various types of tea, like green, black, and oolong, are consumed regularly. Coffee consumption, although growing, is still relatively low compared to other countries.
3. India: India has a strong culture of tea consumption. Chai, which is a spiced milk tea, is a staple beverage across the country. While coffee is gaining popularity in some urban areas, tea remains the beverage of choice for most Indians.
Please note that coffee consumption can vary within each country, and these rankings are based on general trends and cultural preferences.
**Response 2:** There are several countries where people generally drink less coffee compared to Mexico. Some of these countries include:
1. Japan: While coffee has gained popularity in recent years, traditional tea culture remains strong in Japan, resulting in lower coffee consumption overall.
2. China: Despite having a large population, tea is the preferred hot beverage in China, and coffee consumption is relatively low.
3. Saudi Arabia: Traditional Arabic coffee (qahwa) is popular, but in terms of overall coffee consumption, it is lower compared to Mexico.
4. India: Tea is the dominant beverage in India, and coffee consumption is significantly lower in comparison.
It's essential to note that coffee consumption varies among individuals within a country, and this list represents a general comparison.
...

Evaluate these responses.
Select the most consistent response based on majority consensus.
Start your answer with "The most consistent response is Response X" (without quotes).

Figure 7: A full example USC prompt for TruthfulQA. This corresponds to the question in Figure 2b.

