# OpenReview forum: "Universal Self-Consistency for Large Language Models"
_ICML.cc/2024/Workshop/ICL — ICML 2024 Workshop ICL Poster_

### Official Review · Reviewer_G794 · 2024-06-08
**Self-consistency for open-ended generation tasks**

**Rating:** 2
**Fit:** 3
**Confidence:** 3

**Workshop Review:**

One of the approaches to improve the performance of large language models (LLMs) that has shown promising results is self-consistency. The challenge of using self-consistency lies in measuring consistency among a set of predictions from an LLM, especially for open-ended generation tasks . This paper addresses this challenge by proposing an LLM-based approach, which involves asking the LLM to find the most consistent answer. This method effectively identifies consistent outputs across various tasks. The paper is well-written, and the proposed method is applied to different tasks with varying output formats using two different LLMs.

However, there are some limitations to this approach. Firstly, as highlighted in the appendix and section 4.3 of the paper, consistency is not always the most effective method for finding the correct answer. Other approaches, such as those proposed in papers like MCS-SQL [1], suggest that asking LLMs to rank the answers or choose the correct answer could result in higher performance than simply finding the most consistent answer. It would be beneficial to compare consistency-based and ranking-based approaches in this paper, as both can be achieved with a single LLM call. Additionally, for open-ended generation tasks like summarization, the baselines used are only simple greedy decoding and random selection. Stronger baselines could be established by employing heuristic-based methods like using n-grams with thresholds to compute consistency between LLM responses.

[1]: Lee, Dongjun, et al. "MCS-SQL: Leveraging Multiple Prompts and Multiple-Choice Selection For Text-to-SQL Generation." arXiv preprint arXiv:2405.07467 (2024).

**Reason For Not Giving Higher Score:**

1) Lack of comparison with approaches that rank the LLM outputs or choose the correct answer among the generated LLM responses.

2) Lack of strong baselines in cases where standard self-consistency and execution-based self-consistency are not applicable.

3) Including a section on cost versus performance gain with varying numbers of samples would be beneficial.

**Reason For Not Giving Lower Score:**

This paper proposes a novel method to address the challenge of finding the most consistent answer when sampling from LLMs for open-ended generation tasks. They also evaluated their method on various tasks to demonstrate its effectiveness in different scenarios.

---

### Official Review · Reviewer_Pjfj · 2024-06-10

**Rating:** 2
**Fit:** 3
**Confidence:** 2

**Workshop Review:**

The paper addresses a limitation of self-consistency prompting, where the answer has to be in an explicit form / format such that it is easy to extract. The paper proposes to use another LLM to select the most consistent answer from the answer candidates and verify that this can improve performance.

The method makes intuitive sense, but it also has a few limitations. For example, if the output answer consists of large amount of tokens, then feeding all answer candidates into an LLM may exceed the length of context window. Additionally, there is no guarantee that the LLM can actually select the most consistent answer, as it also depends on the instruction-following ability of the LLM. Lastly, the inference speed will be slower than before.

In general, i think the proposed method is a nice trick to improve performance with a few limitations.

**Reason For Not Giving Higher Score:**

The method is quite straightforward with a few obvious limitations.

**Reason For Not Giving Lower Score:**

The experiments show clear improvement with the proposed method.

---

### Meta-Review · Area_Chair_w88W · 2024-06-12

**Recommendation:** 2

**Metareview:**

The paper proposes Universal Self-Consistency (USC), an approach that uses large language models (LLMs) themselves to select the most consistent answer from multiple candidates. This method aims to improve performance on tasks with free-form answers, where traditional self-consistency methods are not applicable.

Both reviewers recognize the novelty and practical improvements presented by USC. Reviewer Pjfj points out potential issues with context length and inference speed, while Reviewer G794 emphasizes the need for comparisons with ranking-based approaches and more robust baselines.

Based on the reviews, the paper meets the threshold for inclusion as a poster in the workshop.

---

### Decision · Program_Chairs · 2024-06-17

**Decision:**

Accept (Poster)

**Comment:**

todo: cite prior relevant work